# Adversarial Perturbations Improve Generalization of Confidence Prediction in Medical Image Segmentation

**Jonathan Lennartz**                                            LENNARTZ@CS.UNI-BONN.DE
**Thomas Schultz**                                              SCHULTZ@CS.UNI-BONN.DE
*University of Bonn, Germany*
*Lamarr Institute for Machine Learning and Artificial Intelligence, Germany*

**Editors:** Accepted for publication at MIDL 2025

## Abstract

Trustworthy methods for medical image segmentation should come with a reliable mechanism to estimate the quality of their results. Training a separate component for confidence prediction is relatively fast, and can easily be adapted to different quality metrics. However, the resulting estimates are usually not sufficiently reliable under domain shifts, for example when images are taken with different devices. We introduce a novel adversarial strategy for training confidence predictors for the widely used U-Net architecture that greatly improves such generalization. It is based on creating adversarial image perturbations, aimed at substantially decreasing segmentation quality, via the gradients of the confidence predictor, leading to images outside of the original training distribution. We observe that these perturbations initially have little effect on segmentation quality. However, including them in the training gradually improves the confidence predictor's understanding of what actually affects segmentation quality when moving outside of the training distribution. On two different medical image segmentation tasks, we demonstrate that this strategy substantially improves estimates of volumetric and surface Dice on out-of-distribution images.

**Keywords:** Failure Detection, Domain Shift.

## 1. Introduction

Algorithms for medical image segmentation always have remaining failure cases, making quality control mandatory (Fournel et al., 2021). Training a neural network that directly predicts segmentation confidence is a simple and computationally efficient solution, and can be adapted to various metrics, differentiable or not. Domain shifts, arising from differences in imaging hardware, patient populations, or acquisition protocols, often derail segmentation models (Guan and Liu, 2022). Unfortunately, in this situation, where error detection is needed the most, direct confidence prediction is the least reliable: Not only does it have to deal with inputs that differ from those it has been trained on; when segmentations degrade markedly, it also needs to extrapolate beyond the training range of its outputs.

In this work, we address these challenges and substantially increase the robustness of direct confidence prediction for medical image segmentation, with the goal of turning this simple and efficient strategy into a practicable solution. We achieve this with a novel approach to training such predictors, augmenting the training data with adversarial examples that are outside of the original distribution and lead to lower segmentation quality. These adversarial examples are derived from the predictor itself, so that including them in the training along with their true effects establishes a feedback loop in which the predictor

learns which perturbations actually affect segmentation quality. We demonstrate that this greatly improves confidence prediction under scanner changes in two real-world cardiac and prostate MRI datasets. Our approach does not require any modification of the underlying segmentation network, and can be adapted to predict different quality metrics.

Our proposed strategy differs substantially from established adversarial training approaches (Bai et al., 2021) in both its goal and implementation. While typical methods focus on increasing robustness against adversarial attacks, our aim is to improve generalization to new domains. This shift necessitates extrapolation beyond the original output distributions—a requirement in our context due to lower segmentation accuracy in new domains compared to the training data. Consequently, we propose a method to generate adversarial examples that reduce segmentation accuracy by a pre-specified amount. Additionally, our approach aligns two networks - one for segmentation and one for confidence prediction - unlike standard adversarial training.

## 2. Related Work

The use of one neural network to directly predict the confidence of another is an active topic of research (Corbière et al., 2019; Fournel et al., 2021; Besnier et al., 2021; Rahman et al., 2022). Similar to the ConfidNet (Corbière et al., 2019), our approach predicts confidence from activation maps. Our main contributions are an extension of that framework to image-level predictions of segmentation quality, and a novel training scheme that includes adversarial perturbations, increasing robustness to domain shifts.

Adversarial perturbations have been shown to be a useful strategy for learning segmentation quality previously (Besnier et al., 2021). However, that previous work made localized changes to street scenes, to improve robustness with respect to unknown objects. Our work is concerned with global changes that arise in medical images when changing acquisition devices or protocols, and therefore derives perturbations in a completely different way, from the confidence predictor itself.

Indirect confidence estimation methods often perform better than direct prediction in out of distribution scenarios. Unfortunately, they can be a computational burden during inference (Valindria et al., 2017) or require specialized architectures, e.g., dropout layers for approximate Bayesian inference (Roy et al., 2018). As part of our experiments, we compare our approach against the latter in terms of quality and computational effort.

## 3. Methodology

### 3.1. Confidence Predictor

We adapt the ConfidNet architecture (Corbière et al., 2019) to the task of per-image confidence prediction for medical image segmentation. Specifically, we attach a confidence predictor $C_\phi$ at the penultimate resolution level of a U-Net $f_\theta$ and train $C_\phi$ to predict $f_\theta$'s true confidence score $g$. To illustrate that $C_\phi$ can be trained to predict both overlap- and boundary-based confidence scores, our experiments include volumetric and surface dice (Maier-Hein et al., 2024) as choices of g. In our main experiments, $f_\theta$ is frozen, so that the original segmentation network remains intact, and $C_\phi$ can re-use the results of its forward pass. In an ablation study, we demonstrate that fine-tuning a copy of $f_\theta$ for the purpose of

---

**Algorithm 1** Adversarial Perturbation Scheme

---

1: **Notation**:
2:     U-Net $f_\theta$, penultimate resolution level features $z_\theta$
3:     Confidence predictor $C_\phi$, true confidence $g : \mathbb{R}^n \times \mathbb{R}^n \to [0,1]$
4:     Loss function $\mathcal{L} : [0,1] \times [0,1] \to \mathbb{R}$
5: Initialize $\theta, \phi, \;\; \alpha \leftarrow 0.8, \;\; \eta \leftarrow 0.01$
6: Initialize ADV_BUFFER$[0] \leftarrow (x_0, y_0) \in \mathcal{D}$
7: **while** not converged **do**
8:     **1. Form a batch of size** $2B$**:**
9:       New input $(x_i, y_i) \in \mathcal{D}$ as clean half $\mathcal{C}$.
10:      $(x'_{i-1}, y_{i-1}) \leftarrow$ ADV_BUFFER$[i]$ as adversarial half $\mathcal{A}$.
11:     **2. Forward Pass:**
12:     **for all** $(x, y) \in \mathcal{C} \cup \mathcal{A}$ **do**
13:       $\hat{y} \leftarrow f_\theta(x), \;\; z \leftarrow z_\theta(x)$
14:       $\hat{s} \leftarrow C_\phi(z), \;\; s \leftarrow g(\hat{y}, y)$
15:     **end for**
16:     $\text{loss} = \displaystyle\sum_{x \in \mathcal{C} \cup \mathcal{A}} \mathcal{L}(\hat{s}, s)$
17:     Update $\phi$ by backpropagating $\nabla_\phi(\text{loss})$
18:     **3. Compute Next Iteration's Adversarial Perturbations:**
19:     **for all** $(x, y) \in \mathcal{C}$ **do**
20:       $\nabla_x \;\leftarrow\; \dfrac{\partial\, C_\phi(z_\theta(x))}{\partial x}$
21:       $\epsilon_{\Delta_s} \leftarrow C_\phi(z) - C_\phi(z') - \Delta_s, \quad \alpha \leftarrow \alpha - \eta \cdot \epsilon_{\Delta_s}$
22:       $\delta \;\leftarrow\; -\alpha \dfrac{\nabla_x}{\|\nabla_x\|^2 + \epsilon}, \;\; x' \leftarrow x + \delta$
23:     **end for**
24:     ADV_BUFFER$[i+1] \leftarrow (x'_i, y_i)$
25: **end while**

---

confidence prediction, as in the ConfidNet, slightly improves confidence prediction further, at an increased computational cost. Details of our architecture are given in Section 3.4.

### 3.2. Learning the Effects of Adversarial Perturbations

After pre-training the confidence predictor $C_\phi$ for 100 batches on the same data, and with the same (non-adversarial) augmentations that were used to train $f_\theta$, we start adding adversarial perturbations, as detailed in Algorithm 1. Our perturbations are based on the negative gradient of the predicted confidence score with respect to the input image, i.e., they represent a change to the image that $C_\phi$ expects to decrease segmentation quality. By processing these adversarial examples with $f_\theta$, and supervising the training of $C_\phi$ with the resulting confidence scores, we establish a feedback loop in which $C_\phi$ learns which deviations from the original training distribution actually affect segmentation quality.

Adversarial perturbations are computed in lines 18–22: In line 21, each image $x$ is modified according to a single gradient step with factor $\alpha$, divided by the squared gradient

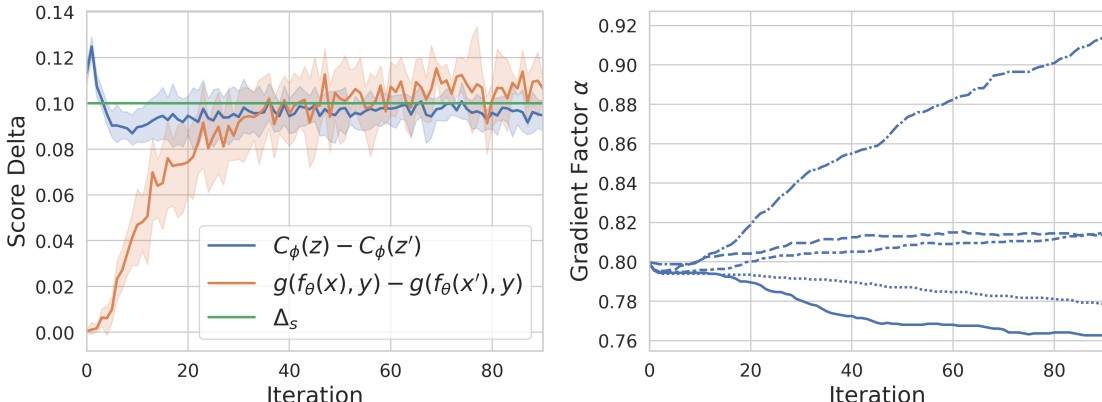

Figure 1: Left: Effects of adversarial perturbations on segmentation network $f_\theta$ and confidence predictor $C_\phi$. Over time, the predictor learns to generate perturbations that actually affect the segmentation. Right: The gradient factor $\alpha$ evolves differently over five runs, illustrating the need to adapt it during training.

norm. In line 20, $\alpha$ is automatically adjusted to reduce the predicted confidence, on average, by a pre-specified amount $\Delta_s$, whose choice is discussed in Section 3.3. Division by the squared gradient norm is motivated by a first-order Taylor expansion of $C_\phi$, where it leads to a constant change in value. A small positive $\epsilon$ guarantees numerical stability.

Figure 1 (left) illustrates our training process for five runs with $\Delta_s = 0.1$. $C_\phi(z) - C_\phi(z')$ is the predicted difference in segmentation quality, based on activations $z$ from the original image and $z'$ from the perturbed one. Adjusting $\alpha$ makes it approximate the desired value $\Delta_s = 0.1$. Interestingly, the actual difference $g(f_\theta(x), y) - g(f_\theta(x'), y)$ between segmentations of the original image $x$ and perturbed image $x'$, as rated by the quality metric $g$ with respect to the ground truth $y$, is very low initially, indicating that the predictor does not yet manage to create effective adversarial perturbations. This shows that it lacks an understanding of which deviations from the input distribution lead to a deterioration of segmentation quality. After a few dozen iterations, our feedback loop successfully aligns the predictor with the actual behavior of the segmentation network. This alignment is further illustrated in Figure 4, which shows examples of perturbations created with or without adversarial training along with their actual effects on the segmentation.

Much of the remaining code in Algorithm 1 is devoted to an efficient implementation of our training scheme, which saves computation by using each image twice, once with, once without adversarial perturbation. This allows us to update the predictor for the current batch and, in the same forward pass, generate adversarial perturbations which are cached in a buffer to be included in the next batch, leading to a 50:50 ratio of original and perturbed images in each batch. For computing weight updates and image perturbations, we can retain the same computation graph to reduce redundant operations and end up with a minimal overhead that integrates well into modern automatic differentiation frameworks.

### 3.3. Learning the Strength of Adversarial Perturbations

We introduce a hyperparameter $\Delta_s$ to control the effect of adversarial perturbations on the predictor, thereby making the process both interpretable and consistent during training. We run an ablation study for $\Delta_s \in \{0.05, 0.1, 0.2\}$ (see Table 3) and find that the framework is rather robust for these settings. For simplicity, we choose $\Delta_s = 0.1$ for all experiments.

Control over the effect is realized by continuously updating $\alpha$ during adversarial perturbation steps. For a sufficiently small $\alpha$, we assume the effect on $C_\phi$ is monotonic and thus use a simple update rule described in line 20 in Algorithm 1. For five runs with similar hyperparameters, we report $\alpha$-values over time in Figure 1 (right). After some iterations, the error in offset $e_{\Delta_s}(z, z')$ stabilizes close to zero, while the gradient factor $\alpha$ continues to evolve over time, illustrating the need to continuously adapt this factor during training.

### 3.4. Implementation Details

Since the original ConfidNet (Corbière et al., 2019) cannot be used for image-level confidence prediction from segmentation features, we propose a simple, wide but shallow, model: Two $3 \times 3$ convolution blocks reduce the number of channels from 64 to 8, followed by two fully connected layers which reduce the remaining feature dimension ($64^2 \times 8$ for M&M and $96^2 \times 8$ for PMRI images) via a hidden dimension of 128 to confidence scores. We train score predictors by attaching them to the penultimate resolution level of a U-Net, which is frozen during training. We use the MSE loss $\mathcal{L}$ between $\hat{s}$ and $s$, the Adam optimizer with a learning rate of $10^{-5}$ and default parameters, and a batch size of 32 that is effectively doubled in step 7 of Algorithm 1. We train all score predictors for at most 100 epochs with 100 batches each and terminate if the validation loss stops improving, with a patience of 20 epochs, retaining the checkpoint from 20 epochs earlier if no improvement is seen. We select volumetric and surface dice as confidence scores $g$ and train a separate predictor for each of them. In multi-class settings, we predict class-wise scores and aggregate later.

## 4. Experimental Setup and Results

### 4.1. Datasets

We employ two MRI datasets to evaluate domain generalization in segmentation tasks. In a preprocessing step, we homogenize voxel-spacings and pad or crop images to a uniform size within each dataset. For both datasets, we use the domain with the most subjects as the source domain and the remaining domains as target domains. The exact number of subjects and slices in each domain is listed in Table 2.

***Heart MRI.*** The second version of the M&Ms Challenge dataset (Campello et al., 2021; Martín-Isla et al., 2023) consists of 8128 annotated cardiac MRI slices across 360 subjects from sites in different countries, acquired using seven different scanning devices, which we use to define our domains. Each image is annotated with three segmentation classes (left ventricle, right ventricle, and myocardium), resulting in a detailed dataset with good support that is well suited for deep learning applications. In our experiments, we refer to it as the M&M dataset.

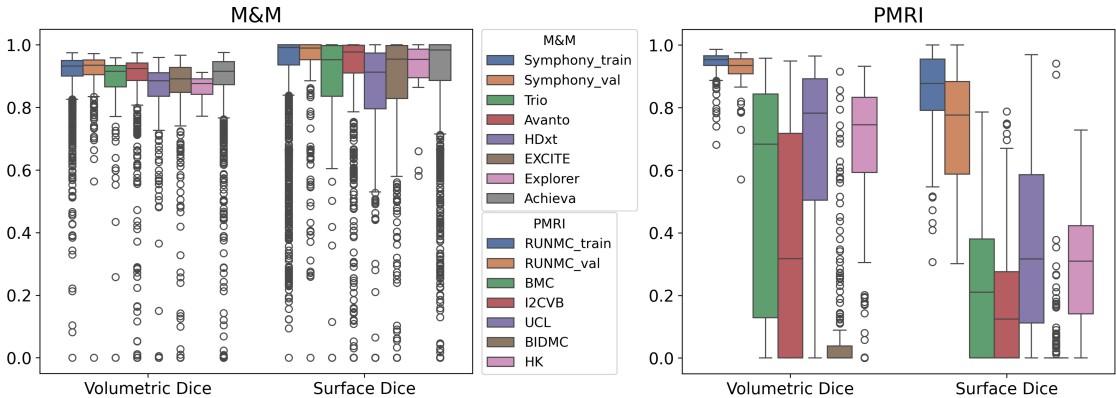

Figure 2: Segmentation quality in terms of volumetric and surface Dice on training and validation data from the source domain, as well as on different target domains. A massive domain shift is observed in PMRI, a more subtle one in M&M.

**Prostate MRI.** (Liu et al., 2020) collect T2-weighted MRI scans of the prostate and their respective binary segmentation masks from six institutions spanning three public datasets. Each institution has distinct imaging devices, protocols and field strengths, resulting in a rich collection of domain shifts with relatively small case numbers. In total, the dataset contains 1773 annotated slices across scans from 116 subjects. In our experiments, we refer to it as the PMRI dataset.

### 4.2. Image Segmentation with U-Nets

To reduce the impact of confounding factors as much as possible, we use the same U-Net architecture and confidence predictor across datasets and tasks. We use MONAI (Cardoso et al., 2022) to implement a U-Net with 32 initial channels and a depth of four with four residual units per level. Furthermore, we use dropout with a dropout rate of 0.1 per ADN layer to ensure a fair comparison to a previous described indirect confidence estimation technique (Roy et al., 2018). We train the U-Net with a mixed loss of Dice and cross-entropy, using the Adam optimizer with a learning rate of $10^{-3}$ and default settings. We copy the default data augmentations from the nnU-Net (Isensee et al., 2021) and train with learning rate scheduling and early stopping based on a held-out validation set. Albeit following standard strategies to address potential domain shifts, we still observe a moderate drop in performance on M&M and essentially a complete shift in confidence score distribution for PMRI, see Figure 2. In combination with the diverse dataset characteristics and our two confidence scores, we hope to provide a convincing test bed for confidence prediction in medical image segmentation.

### 4.3. Confidence Prediction

The goal of our proposed adversarial perturbation scheme is to improve the confidence predictor's generalization capabilities. To quantify whether this goal has been met, we report Pearson correlation, excess area under the risk-coverage-curve (eAURC) (Geifman et al., 2018) and mean absolute error for volumetric and surface dice scores, specifying mean and standard deviation across five runs.

In Figure 3, we compare our proposed framework for direct confidence prediction, with and without our adversarial perturbation scheme, to an approximate Bayesian method that we refer to as score agreement (Roy et al., 2018), and that has been identified as a robust baseline for failure detection in a recent comparative benchmark (Zenk et al., 2025). It is based on taking Monte Carlo samples of segmentation masks with test-time dropout, and measuring the agreement between them by averaging pairwise quality metrics, in our case, volumetric or surface Dice. Following the same setup as the authors, we take $N = 15$ samples to saturate performance, resulting in 105 pairwise comparisons. This procedure results in a confidence score that correlates to segmentation accuracy, but does not estimate the corresponding quality metric directly. Therefore, it does not make sense to compute mean absolute error for it.

On most domains, for both datasets and confidence measures, our adversarial perturbation scheme significantly improves the predictor's ability to generalize in terms of achieving higher correlation, lower eAURC, and lower mean absolute error. Our approach also narrows the gap to the computationally much more expensive score agreement methodology, even outperforming it in several cases. Additionally, we run score agreement with only $N = 2$ samples, to make its computational complexity more comparable to direct confidence prediction. In this scenario, direct prediction is mostly superior.

We also explored average aggregated predictive entropy over $N = 15$ dropout samples as another baseline, but found its results too weak to include them in the Figure. On average over datasets and confidence metrics, it achieved a Pearson correlation below 0.3.

In addition to the comparisons in Figure 3, we also investigate benefits of fine-tuning the segmentation network's backbone alongside the confidence predictor but find only marginal improvements, at the cost of increased computational and implementation complexity, see Table 3.

### 4.4. Computation Times

Table 1 compares the running times of our proposed approach (top row) to those of computing score agreement. Due to its shallow architecture, our confidence predictor adds negligible overhead to the segmentation network itself. Fine-tuning (second row) roughly doubles our running time, with marginal benefits reported in Table 3.

In any case, our running times are much shorter than for the 15 forward passes that are required to compute saturated score agreement and still considerably shorter than score agreement with two forward passes. Most importantly, inference times in our approach are independent of the computational complexity of the selected quality metric, which by far dominates the overall running time of score agreement.

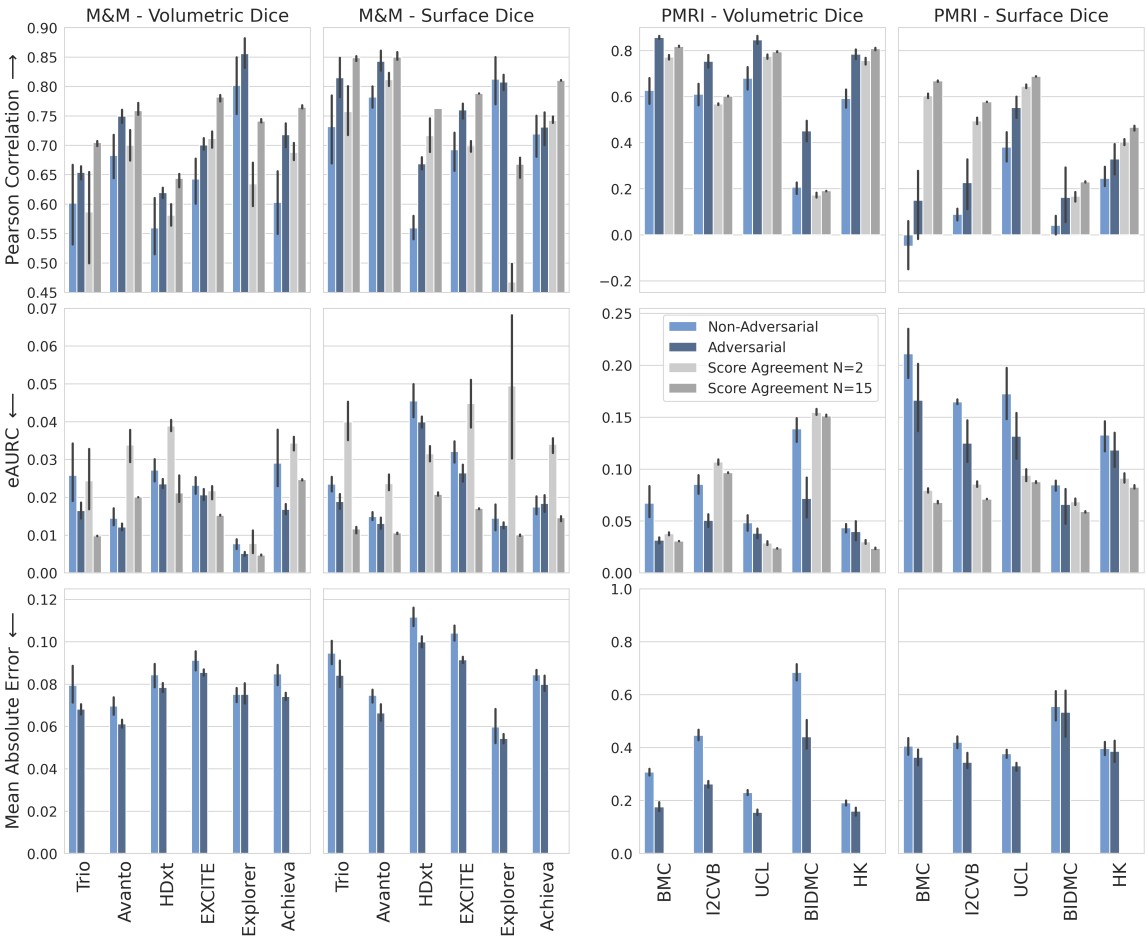

Figure 3: Evaluation of our proposed direct confidence prediction. Adversarial perturbations increase correlation, and decrease excess area under the risk-coverage-curve, as well as mean absolute error, in almost all cases. In several cases, it even provides better correlation and eAURC than the computationally much more expensive score agreement approach, which does not provide absolute predictions.

## 5. Discussion

In this work, we presented an adversarial perturbation scheme that strengthens direct confidence prediction in medical image segmentation under domain shifts. It is based on the idea that the alignment between predicted and actual accuracy of a segmentation model on out-of-distribution data can be improved by learning the effects of adversarial perturbations. At the same time, our training scheme widens the value range of quality metrics that are observed during training, and thus facilitates prediction of scores that are lower than those from the original training images.

| Method | M&M | PMRI |
|---|---|---|
| $f_\theta + C_\phi$ | 0.0009 | 0.0020 |
| $+ f_\theta$ after fine-tuning | 0.0018 | 0.0038 |
| Volumetric Dice Agreement (N=2) | 0.0059 | 0.0062 |
| Volumetric Dice Agreement (N=15) | 0.1011 | 0.0802 |
| Surface Dice Agreement (N=2) | 0.0230 | 0.0153 |
| Surface Dice Agreement (N=15) | 1.8964 | 1.0683 |

Table 1: Inference times in seconds per image, averaged across 100 runs on a single NVIDIA A40 GPU. We use MONAI's metric implementations and calculate agreements in a single batch. Our proposed approach (top row) is much faster than score agreement (row two and below), especially with expensive quality metrics such as surface Dice.

Our approach does not require any changes to the underlying segmentation network and has negligible computational overhead during inference. Moreover, we describe an efficient algorithm for training, re-using computations that are anyway required for training the predictor to generate adversarial perturbations.

On two MRI datasets and for an overlap- and a boundary-based quality score, our adversarial scheme improves the confidence prediction baseline across all metrics. In terms of correlation and eAURC, which do not require absolute estimates, it narrows the gap to the more computation-heavy score agreement method (Roy et al., 2018) and sometimes even surpasses it.

While the relative improvement from our contribution is evident, it does not yet establish a new benchmark for failure detection, and we observe a limitation with respect to the remaining absolute errors on the PMRI dataset (Figure 3). The fact that they are still substantially larger than in M&M is unsurprising, given that the degradations that our adversarial perturbations aim for ($\Delta_s = 0.1$) are far lower than the real shift we observe in the data (see Figure 2). Extending our framework to include more severe perturbations is thus an obvious goal for future work. It is clear from our ablation that this cannot simply be achieved by scaling up $\Delta_s$, but will require a procedure that is more complex than taking a single gradient step.

In summary, we believe that our work demonstrates the potential of suitably trained direct confidence prediction, even in cases of domain shift. Considering that inference time is up to three orders of magnitude faster than for score agreement, and that we obtain absolute estimates of segmentation quality, which score agreement cannot provide, we consider further refinement of this approach to be a worthwhile goal of future research.

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

## Appendix A. Data Splits

| | M&M Dataset | | | | | |
|---|---|---|---|---|---|---|
| | Sym. (train/val) | Trio | Avanto | HDxt | Excite | Explorer | Achieva |
| Cases | 172 | 5 | 42 | 25 | 27 | 1 | 88 |
| Slices | 2699/299 | 94 | 695 | 426 | 459 | 18 | 1422 |

| | PMRI Dataset | | | | | |
|---|---|---|---|---|---|---|
| | RUNMC (train/val) | BMC | I2CVB | UCL | BIDMC | HK | |
| Cases | 30 | 30 | 19 | 13 | 12 | 12 | |
| Slices | 378/41 | 324 | 505 | 171 | 197 | 157 | |

Table 2: Total slice counts for the M&M and PMRI datasets. Training and validation splits are consistent across U-Net and predictor trainings.

## Appendix B. Gradient Visualization

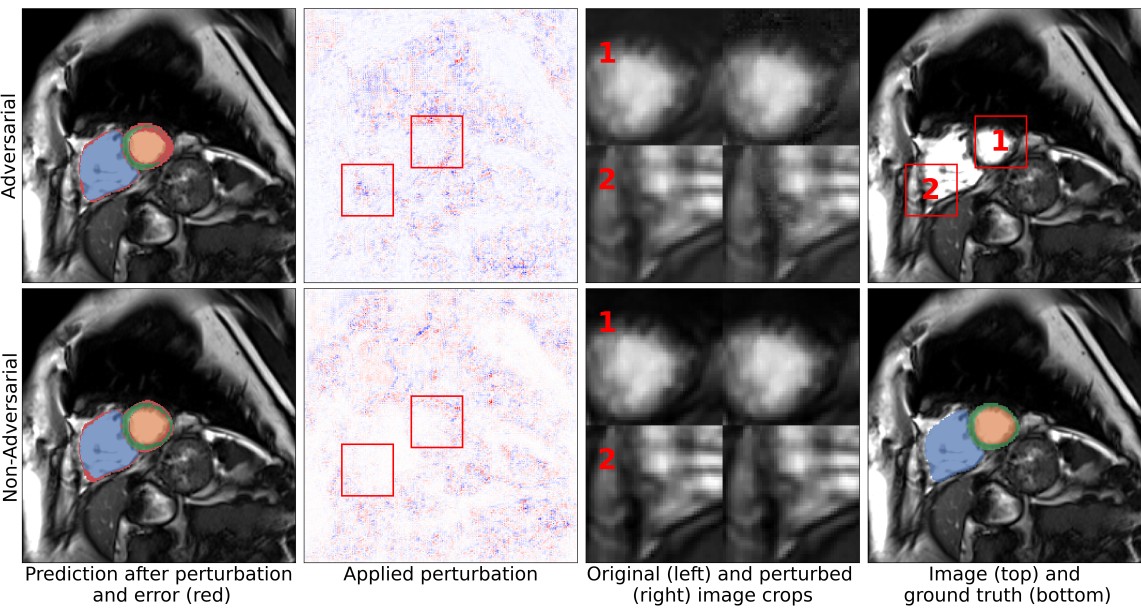

Figure 4: Example from the M&M dataset comparing the behavior of confidence predictors trained with and without our adversarial perturbation scheme on volumetric dice prediction. In both cases, adversarial perturbations affect the whole image, but are difficult to visually discern. Both perturbations decrease predicted confidence (0.94 → 0.87 top, 0.92 → 0.70 bottom). However, only the perturbation from our method actually results in a degradation of the segmentation quality (0.90 → 0.81), while the one from a predictor without adversarial training has little effect on the segmentation (0.90 → 0.90).

## Appendix C. Ablation Study

| | Pearson Correlation | | | | | |
|---|---|---|---|---|---|---|
| Method | Achieva | Avanto | EXCITE | Explorer | HDxt | Trio |
| $\Delta_s$ | | | | | | |
| 0.05 | $.675 \pm .018$ | $.736 \pm .018$ | $.662 \pm .032$ | $.847 \pm .019$ | $.608 \pm .035$ | $.624 \pm .035$ |
| 0.1 | $.718 \pm .024$ | $.749 \pm .015$ | $.700 \pm .013$ | $.856 \pm .033$ | $.619 \pm .012$ | $.654 \pm .015$ |
| fine-tuned | $.730 \pm .033$ | $.754 \pm .013$ | $.716 \pm .017$ | $.858 \pm .036$ | $.632 \pm .014$ | $.668 \pm .011$ |
| 0.2 | $.719 \pm .037$ | $.779 \pm .023$ | $.695 \pm .016$ | $.850 \pm .041$ | $.631 \pm .019$ | $.616 \pm .044$ |
| | eAURC | | | | | |
| 0.05 | $.019 \pm .003$ | $.012 \pm .001$ | $.023 \pm .003$ | $.004 \pm .001$ | $.025 \pm .002$ | $.018 \pm .002$ |
| 0.1 | $.017 \pm .002$ | $.012 \pm .001$ | $.021 \pm .002$ | $.005 \pm .001$ | $.024 \pm .002$ | $.016 \pm .003$ |
| fine-tuned | $.016 \pm .002$ | $.012 \pm .001$ | $.020 \pm .002$ | $.005 \pm .001$ | $.023 \pm .002$ | $.015 \pm .001$ |
| 0.2 | $.017 \pm .001$ | $.012 \pm .001$ | $.022 \pm .002$ | $.007 \pm .004$ | $.023 \pm .002$ | $.017 \pm .001$ |
| | MAE | | | | | |
| 0.05 | $.076 \pm .002$ | $.064 \pm .004$ | $.088 \pm .004$ | $.078 \pm .005$ | $.078 \pm .006$ | $.071 \pm .002$ |
| 0.1 | $.074 \pm .002$ | $.061 \pm .002$ | $.086 \pm .002$ | $.075 \pm .006$ | $.078 \pm .003$ | $.068 \pm .003$ |
| fine-tuned | $.075 \pm .003$ | $.061 \pm .002$ | $.086 \pm .002$ | $.074 \pm .008$ | $.076 \pm .004$ | $.074 \pm .004$ |
| 0.2 | $.073 \pm .003$ | $.060 \pm .002$ | $.083 \pm .003$ | $.075 \pm .006$ | $.077 \pm .004$ | $.067 \pm .003$ |

Table 3: Ablation exploring different values of $\Delta_s$, as well as fine-tuning of the segmentation network's parameters for confidence prediction (with $\Delta_s = 0.1$) for volumetric Dice prediction on test domains of the M&M dataset. Correlation and eAURC improve for our default setting $\Delta_s = 0.1$ compared to a reduced $\Delta_s = 0.05$, but increasing it to $\Delta_s = 0.2$ does not always yield further improvement. Overall, results are stable with respect to variations in $\Delta_s$. Fine-tuning leads to a small benefit in correlation and eAURC, but also to higher complexity and running times. MAE benefits from larger $\Delta_s$, but less clearly from fine-tuning.

