# OpenReview forum: "Adversarial Perturbations Improve Generalization of Confidence Prediction in Medical Image Segmentation"
_MIDL.io/2025/Conference — MIDL 2025 Poster_

### Official Review · Reviewer_BthZ · 2025-02-19

**Confidence:** 4
**Preliminary Rating:** 4
**Recommendation:** Oral
**Final Rating:** 3

**Summary:**

Unless there is in place a method which tells a user how good an automatically generated segmentation is, visual inspection conducted by an expert is essential for the deployment of such system.
With this in mind, this paper presents a clever way to improve estimation of a model's predictive confidence. Training a model to estimate another model's confidence has its limitations especially when it comes to using it under data domain shifts - which is common with imaging data. This manuscript shows that by adversarially adding perturbations to the input data during training, a simultaneously trained confidence predictor becomes aware of what really makes a segmentation good or bad, making the predictor reliable also on shifting domains.
This happens because the confidence predictor head trainer and the segmentation head trained enter in a loop in which the confidence predictor arguably understands what affects the segmentator.

**Strengths:**

The paper is well written and explain the used algorithm in details motivating choice of each parameter.

Some strengths about the proposed approach:
- it improves robustness of confidence prediction;
- it improves estimates of the commonly used volumetric and surface DICE metrics as well as (potentially) any other metric;
- it can be added to any model training, with minor computing overhead.

**Weaknesses:**

Additional weaknesses/doubts are reported in the "Questions To Address In The Rebuttal" section.
Some weaknesses:
- In my opinion, the community would greatly benefit from open-sourcing this algorithm. From the manuscript it seems that adapting to different imaging modality/scenarios should be simple, making it almost plug and play. Are the authors planning to open source their code?

**Detailed Comments:**

Please refer to the other sections for all the comments.

**Justification Of The Final Rating:**

The authors’ response to the reviewers’ comments appears to be minimal, and at times, somewhat defensive. Additionally, while the reviewers encouraged open-sourcing the implementation earlier to facilitate a better understanding of the algorithm, the authors have stated they will do so only upon acceptance. Given these aspects, the paper is borderline in its current form.

**Justification Of The Preliminary Rating:**

This paper is very relevant to MIDL. It proposes a novel approach to estimating the quality of a model's segmentation output and we all need a 1 working solution!
Experiments on two datasets were conducted and the paper is very well written. While I really support this paper, I believe some clarifications are needed for me to be able to make a final rating.

**Questions To Address In The Rebuttal:**

Curiosity about the perturbations visual appearance and their effect:

- The paper is about perturbing images, so visualizing such perturbations would make the reader appreciate the work more. In my opinion, it would be really interesting to observe how perturbed images evolve during training along with the quality metric and predicted value. I believe this could complement the information provided in Figure 1 left.

Curiosity about the adversarial perturbations:
- How do they look like? Is the adversarial gradient merely adding noise to an image, or did the authors observe actual imaging artefacts?
- Instead of just using adversarial gradients, could the addition of physics informed artefacts make the approach more robust and applicable to shifts observed in real world scenarios (where the shift is not just between acquisitions or scanner brands)?
- Do you expect the approach to work as it is without expecting major tunings on other image modalities (e.g. CT, ultrasounds, x-ray, OCT, retinal images, etc) but just retraining on a new use case?

Curiousity about training of the confidence predictor:
- In section 3.3 the authors explain that alpha continues to evolve over time, does this make training unstable? Have they identified some ideal initialization value for alpha which might work for various imaging type?
- The authors stop training if the validation loss stops improving. For how many additional iterations do they attempt training before training stops?
- In multi-class segmentation does the predictor computes an average metric or is it per class? have they authors observed whether the approach is equally reliable in smaller/harder to segment structures?

**Special Issue:**

Yes

---

> ### Author Response · Authors · 2025-03-07
>
> Thank you for your detailed and constructive feedback. We appreciate your positive comments regarding our algorithm description and the clarity of our presentation.
>
> Answers to questions to address in the rebuttal:
>
> **Q:** The paper is about perturbing images, so visualizing such perturbations would make the reader appreciate the work more. In my opinion, it would be really interesting to observe how perturbed images evolve during training along with the quality metric and predicted value. I believe this could complement the information provided in Figure 1 left. How do they look like? Is the adversarial gradient merely adding noise to an image, or did the authors observe actual imaging artefacts?”
> **A:** We have added visualizations of perturbations and their effects in Appendix B. Rather than showing an evolution over time—which could be confounded by incomplete training early on—we compare perturbations from fully trained predictors, with and without our adversarial scheme. The visual impression in either case is similar to adversarial examples that have been demonstrated in other contexts. The important point appears to be that such perturbations similarly affect the segmentation and confidence prediction networks, rather than emergence of semantically meaningful image modifications.
>
> **Q:** “Instead of just using adversarial gradients, could the addition of physics informed artefacts make the approach more robust and applicable to shifts observed in real world scenarios (where the shift is not just between acquisitions or scanner brands)?"
> **A:** We agree that this is a potential direction for future work. On the other hand, if such artefacts are to be expected in a real world scenario, one might consider including them as augmentations in the first place, to make the segmentation itself as robust as possible. Our approach is compatible with such augmentation strategies. Our experiments use the same augmentations we used when training the U-Nets (i.e., defaults from nnU-Net).
>
> **Q:** "Do you expect the approach to work as it is without expecting major tunings on other image modalities (e.g. CT, ultrasounds, x-ray, OCT, retinal images, etc) but just retraining on a new use case?”
> **A:** We think so, yes. Our approach is not tailored to any specific image modality. The datasets have been chosen because they allow us to investigate real-world domain shifts, not because of the modality.
>
> **Q:** “In section 3.3 the authors explain that alpha continues to evolve over time, does this make training unstable? Have they identified some ideal initialization value for alpha which might work for various imaging type?"
> **A:** We modified Algorithm 1 to clarify that we initialized alpha at 0.8 in all cases. We found that the automated adaptation, combined with gradient normalization, made it unnecessary to manually tune that hyperparameter. We did not encounter any cases in which it would have led to unstable training.
>
> **Q:** "The authors stop training if the validation loss stops improving. For how many additional iterations do they attempt training before training stops?"
> **A:** We modified 3.4 to clarify that we run a maximum of 100 epochs (with 100 batches each) and use early stopping with a 20-epoch patience, retaining the checkpoint from 20 epochs earlier if no improvement is seen.
>
> **Q:** "In multi-class segmentation does the predictor computes an average metric or is it per class?"
> **A:** We modified 3.4 to clarify that we predict class-wise dice scores and aggregate later.
>
> **Q:** "Have they authors observed whether the approach is equally reliable in smaller/harder to segment structures?”
> **A:** We agree that this is an interesting question for future work, but believe that our current datasets are not ideal for answering it.
>
> Answers to other questions:
>
> **Q:** “The community would greatly benefit from open-sourcing this algorithm. From the manuscript it seems that adapting to different imaging modality/scenarios should be simple, making it almost plug and play. Are the authors planning to open source their code?”
> **A:** We plan to release our code upon acceptance, as we have done for previous projects.
>
> We appreciate the opportunity to clarify these points and thank you again for your supportive comments.

---

> > ### Comment · Reviewer_BthZ · 2025-03-12
> >
> > Thanks for the rebuttal. I noted that also other reviewers brought up whether code could be open sourced.
> >
> > I would like to clarify that reviewers take the review process very seriously and confidentially, so having access to a repo can only benefit their understanding of the idea, since often a good line of code can explain as much or even more than an entire book chapter would.
> >
> > From a reviewer perspective if the code is shared after the paper has been published makes little difference.
> >
> > Please feel free to open source the code or not, but in general it is a good exercise / practice which can positively impact one’s career or even an entire field of research.

---

> > > ### Author Response · Authors · 2025-03-12
> > >
> > > Thank you for the note, the advice is appreciated. While we will open source the Python code to support reproducibility, our goal is to make the paper sufficiently self-contained that a comprehensive understanding of our idea can be gained from it alone. If there are any remaining open points regarding the understanding of our idea, please let us know so that we can clarify them accordingly.

---

### Official Review · Reviewer_nodi · 2025-02-21

**Confidence:** 4
**Preliminary Rating:** 2
**Final Rating:** 3

**Summary:**

The paper proposes to adapt ConfidNet (originally developed for a classification task) to a segmentation task. The aim is to add a confidence score of the segmentation. For that, the authors modify the original architecture and adapt it to an U-Net architecture used for the segmentation. They also propose to learn the confidence predictor with adversarial perturbations (using the gradient of this predictor) to increase the robustness of this predictor. The method is evaluated on 2 MRI datasets: heart and prostate. The data are acquired in different countries with different devices and protocols. The authors used the domain (country/device) with  the largest number of samples to train the model and evaluate it on the other domains: the domain shift is expected to decrease performance and increase model uncertainty. The confidence prediction is evaluated with three metrics : person correlation, eAURC and mean absolute error. The results show that the proposed is faster than the Bayesian state-of-the-art method, but its performance is inferior in most cases. The addition of adversarial perturbations increases the performances compared to the model trained without these perturbations.

**Strengths:**

- The paper addresses an important issue in medical image segmentation: confidence.
- Adding data augmentation based on adversarial perturbations of the confidence predictor improves the performances . The idea is promising and could also be used to train the segmentation network and improve its generalization.

**Weaknesses:**

- The performances of the proposed confidence prediction are low compared to the Bayesian method. In most cases, the computation times of the Bayesian method (<2s in the worst cases) is not a problem in real scenarios. In addition, this time could be reduced by reducing the number of forward passes. Have the authors evaluated for how many forward passes (and so computation time) the performances of the Bayesian method are equivalent to the proposed method?
- For segmentation task, most state-of-the-art methods propose a pixel-level confidence score, indicating which regions are less certain. A global score can also be computed with these methods by aggregating the score of each pixel. The proposed method only provides a global score of the segmentation quality which is less precise. Moreover, the authors do not compare their method with pixel-level methods but only to a very simple method (Bayesian). Perhaps the gradient computed in line 19 of Algorithm 1 could be used to obtain a local information with the proposed method.
- Some details are missing in the paper to fully understand the method and make it reproducible.

**Detailed Comments:**

- Make the code available will improve the reproducibility of the method
- In Algorithm 1, all the notations are not explained.
- Details on g are missing  in implementation details. Please specify the function used (surface dice and volumetric dice ?). Do the experiments with the volumetric and surface dice correspond to two different trainings of the confidence predictor (each with the corresponding metric used to train the network))?
- Details on the training, validation and test set are lacking: number of samples, domain used.

**Justification Of The Final Rating:**

The proposed method seems promising as it improves the performances compared to the baseline. The study conducted on domain shift is also very interesting and important for the community. The manuscript is now very clear. However, as the authors concede, the method is not yet sufficiently developed to surpass the state of the art.

**Justification Of The Preliminary Rating:**

The proposed method offers no improvement over the state of the art, with the exception of computation time, which seems irrelevant for clinical applications...

(see weaknesses and detailed Comments)

**Questions To Address In The Rebuttal:**

- Details should be added (and ideally code shared) to clearly understand the method and its implementation.
- The authors should justified is which scenario computational time is so crucial that we can accept a loss in performance.

**Special Issue:**

No

---

> ### Author Response · Authors · 2025-03-07
>
> Thank you for your detailed and constructive feedback.
>
> Answers to questions to address in the rebuttal:
>
> **Q:** Details should be added (and ideally code shared) to clearly understand the method and its implementation.
> **A:** We added further details. In particular, we added a definition of $\mathcal{L}$ in the Notation section of Algorithm 1 as well as in Section 3.4. Section 3.4 now clarifies that we trained separate networks for volumetric and surface dice. Appendix A now presents dataset information, including train, validation, and test splits. We plan to release our code upon acceptance, as we have done for previous projects.
>
> **Q:** The authors should justified is which scenario computational time is so crucial that we can accept a loss in performance.
> **A:** We modified our discussion (Section 5) to clarify that we do not consider our approach to establish a new benchmark, and to highlight the value that we see in our work despite that. In our opinion, the computational effort and the inability to provide absolute estimates of segmentation quality are fundamental limitations of the score agreement approach. We expect that the remaining gap in eAURC in some (not all) cases is a limitation of the direct prediction strategy that can be closed by follow-up works, which we hope to inspire.
>
> Answers to other questions:
>
> **Q:** "In most cases, the computation times of the Bayesian method (<2s in the worst cases) is not a problem in real scenarios. In addition, this time could be reduced by reducing the number of forward passes. Have the authors evaluated for how many forward passes (and so computation time) the performances of the Bayesian method are equivalent to the proposed method?"
> **A:** Thank you for raising this interesting point. We had selected N=15 based on a recommendation in the reference work. As an additional baseline, we now included an experiment with the smallest possible computational budget (N=2). We found that it is still slower than our approach, and produces worse results in most cases.
>
> **Q:** "For segmentation tasks, most state-of-the-art methods propose a pixel-level confidence score [...]"
> **A:** Image level failure detection is relevant in many cases in which a choice has to be made to either use a segmentation result or to exclude it from downstream analysis. For this task, a recent comparative benchmark (Zenk et al., MedIA 2025, 10.1016/j.media.2024.103392) found score agreement to work extremely well. This agrees with our own experiments with aggregated pixel-level confidence scores, which did not correlate well with surface or volumetric dice and are now briefly included in Section 4.3. If you could provide a specific reference that shows how such a strategy can produce state-of-the-art results for image level failure detection, that would be highly appreciated.
>
> Thank you again for your insightful comments. We appreciate your time and consideration.

---

> > ### Comment · Reviewer_nodi · 2025-03-14
> > **On pixel-level confidence score methods**
> >
> > Thank for the modifications which improve the manuscript.
> >
> > Regarding the last point, with pixel-level confidence score methods, we can go further than a binary classification (accept or reject a segmentation) as a part of the segmentation can be very good, which is not not perceptible with a method that only gives a score. Maybe the context of failure detection can be highlighted in the introduction. The added comparison with aggregated pixel-level confidence method and the new reference reinforce the message.

---

> > ### Author Response · Authors · 2025-03-15
> >
> > Thank you for your feedback. We appreciate your positive remarks on the added comparison and new reference, which we believe have strengthened the overall narrative of our manuscript.

---

### Official Review · Reviewer_p5Qg · 2025-02-22

**Confidence:** 4
**Preliminary Rating:** 3

**Summary:**

This paper introduced an method to improve generization of medical image segmentation model by trainning a confidence predictor using adversarial training strategy.  The approach involves generating adversarial perturbations designed to reduce segmentation quality, using the gradients of the confidence predictor attached to a U-Net architecture.

**Strengths:**

1. The experiment results show that this method can actually improve the performance of model on various metrics.
2. This method can improve the performance of the model on out-of-distribution data, which makes it a possible method to forster the deployment of models in real-world scenerio.

**Weaknesses:**

1. The adversarial training strategy and U--Net architecture are both well-studied areas in AI, which decreases the novety of the paper.
2. The experiment results are good but the paper only conducts experiment on 2 dataset.

**Detailed Comments:**

1. Additional visualizations of adversarial perturbations and their effect on segmentation quality would enhance understanding.
2. A deeper comparison with other confidence estimation methods, beyond score agreement, would strengthen the paper’s positioning.

**Justification Of The Preliminary Rating:**

The clear and well-executed methodology, combined with some empirical validation, supports the paper’s contributions. However, the rating also accounts for the limitations observed in novety and the potential for further exploration in architectural and training variations. Overall, the paper makes contribution but leaves some open questions that needs further investigation.

**Questions To Address In The Rebuttal:**

1. Could the authors state the novety of this paper other than using existing methods?
2. Will this method work on a more complicated model structure or other model structure?

---

> ### Author Response · Authors · 2025-03-07
>
> Thank you for your constructive feedback. We appreciate your positive remarks regarding our experimental results and the potential for real-world application.
>
> Answers to questions to address in the rebuttal:
>
> **Q:** Could the authors state the novelty of this paper other than using existing methods?
> **A:** We highlight the novelty of our paper in a new paragraph at the end of the introduction, emphasizing that our proposed approach differs significantly from previously studied adversarial training strategies, both in terms of its goals and its implementation.
>
> **Q:** Will this method work on a more complicated model structure or other model structure?
> **A:** Our current experiments focus on the U-Net, since it is by far the most widely used architecture for medical image segmentation. However, our main idea—establishing an adversarial feedback loop based on differentiable confidence prediction—should carry over to other model structures. We would be curious to hear which specific models you are most interested in.
>
> Answers to other questions:
>
> **Q:** Additional visualizations of adversarial perturbations and their effect on segmentation quality would enhance understanding.
> **A:** We have now added visualizations of our perturbations along with their corresponding effects in Appendix B.
>
> **Q:** A deeper comparison with other confidence estimation methods, beyond score agreement, would strengthen the paper’s positioning.
> **A:** In Section 4.3, we now clarify that score agreement has been chosen because it has been recommended in a recent comparative benchmark of failure detection methods (Zenk et al., MedIA 2025, 10.1016/j.media.2024.103392). For a deeper comparison, we added a variant with fewer samples, that better matches the computational effort of our proposed method. In the text, we now also briefly report experiments with average per-pixel entropy as another baseline method. However, it correlates so poorly with Dice and Surface Dice that we decided not to include it in the (already rather busy) Figure 3. Similarly, we tried Reverse Classification Accuracy, but found that it took so much time (>7 mins / image) that including it as another baseline in all experiments was infeasible.
>
> Thank you again for your valuable insights. We are grateful for the opportunity to refine our work and look forward to further discussions.

---

### Author Rebuttal · Authors · 2025-03-07

**Rebuttal:**

Dear reviewers,

thank you again for your detailed and constructive feedback. Please find attached our preliminary revision with relevant changes marked in red.

We are looking forward to the discussion!

**Supporting Material:**

/attachment/6b40d5c1f5331c0965c83df18d89ce4c19952119.zip

---

### Comment · Program_Chairs · 2025-03-17
**Forwarding a message from authors to AC**

Dear AC,

We're forwarding a message from the authors (see below), who have raised concerns about one of the reviews. Please have a close look, and note that your decision notification should not be based on code availability.

Forwarded message:

Dear paper co-chairs,

I am writing to share our experience with the MIDL discussion phase of
submission 190. Reviewer Bthz used the discussion to pressure us to give
them access to the implementation of our approach for which, given the
preliminary numerical ratings, it seemed uncertain whether it would get
accepted, saying that they "take the review process very seriously and
confidentially" and that it would "benefit their understanding of the idea".

We asked what specific point was still unclear and required inspecting
our code, despite the fact that we had carefully revised our manuscript
and answered all individual reviewer questions as part of the rebuttal.
Instead of providing an answer, the reviewer lowered their numerical
rating, citing the fact that we are only willing to publish code upon
acceptance as the primary reason. I believe that, considering that we
added multiple new experiments, references, and about a page of new text
during the rebuttal, calling our response to the reviewers' comments
"minimal" is not justified.

It is my impression that the MIDL discussion phase did not help to
achieve a more fair assessment, but rather strengthened the reviewers'
position of power to a point where it invites the abuse of that power.
If possible, I would like to hear your thoughts about this.

Best regards,

   Thomas Schultz

---

### Meta-Review · Area_Chair_UBYT · 2025-03-21

**Recommendation:** Accept (Poster)
**Confidence:** 3

**Metareview:**

This paper introduced an method to improve generization of medical image segmentation model by trainning a confidence predictor using adversarial training strategy. The approach involves generating adversarial perturbations designed to reduce segmentation quality, using the gradients of the confidence predictor attached to a U-Net architecture.
Proposed method seems promising as it improves the performances compared to the baseline. The study conducted on domain shift is also very interesting and important for the community. The manuscript is clear.
Even if the method is not yet sufficiently developed to surpass the state of the art, it could be interesting to see this approach presented in MIDL.